
# New Particle Formation Events Detection with Deep Learning

Peifeng Su[1,2], Jorma Joutsensaari[3], Lubna Dada[4,5], Martha Arbayani Zaidan[2,6], Tuomo Nieminen[2,7], Xinyang Li[2], Yusheng Wu[2], Stefano Decesari[8], Sasu Tarkoma[9], Tuukka Petäjä[2,6], Markku Kulmala[2,6], and Petri Pellikka[1,2]

[1]Department of Geosciences and Geography, University of Helsinki, FI-00014 Helsinki, Finland
[2]Institute for Atmospheric and Earth System Research (INAR/Physics), Faculty of Science, University of Helsinki, FI-00014 Helsinki, Finland
[3]Department of Applied Physics, University of Eastern Finland, P.O. Box 1627, 70211 Kuopio, Finland
[4]Extreme Environments Research Laboratory, Ecole Polytechnique Fédérale de Lausanne (EPFL) Valais, Sion, 1951, Switzerland
[5]Laboratory of Atmospheric Chemistry, Paul Scherrer Institute, Villigen, 5232, Switzerland
[6]Joint International Research Laboratory of Atmospheric and Earth System Sciences, School of Atmospheric Sciences, Nanjing University, Nanjing 210023, China
[7]Institute for Atmospheric and Earth System Research (INAR/Forest Sciences), Faculty of Agriculture and Forestry, University of Helsinki, FI-00014 Helsinki, Finland
[8]Italian National Research Council Institute of Atmospheric Sciences and Climate (CNR-ISAC), Bologna 40129, Italy
[9]Department of Computer Science, Faculty of Science, University of Helsinki, FI-00014 Helsinki, Finland

**Correspondence:** Peifeng Su (peifeng.su@helsinki.fi)

**Abstract.** Atmospheric new particle formation (NPF) is an important source of climate-relevant aerosol particles which has been observed at many locations globally. To study this phenomenon, the first step is to identify whether an NPF event occurs or not on a given day. In practice, NPF event identification is performed visually by classifying the NPF event or non-event days from the particle number size distribution surface plots. Unfortunately, this day-by-day visual classification is time-consuming, labor-intensive, and the identification process renders subjective results. To detect NPF events automatically, we regard the visual signature (banana shape) which has been observed all over the world in NPF surface plots as a special kind of object, and a deep learning model called Mask R-CNN is applied to localize the spatial layouts of NPF events in their surface plots. Utilizing only 358 human-annotated masks on data from the Station for Measuring Ecosystem and Atmospheric Relations (SMEAR) II station (Hyytiälä, Finland), the Mask R-CNN model was successfully generalized for three SMEAR stations in Finland and the San Pietro Capofiume (SPC) station in Italy. In addition to the detection of NPF events (especially the strongest events), the presented method can determine the growth rates, start times, and end times for NPF events automatically. The automatically determined growth rates agree with the growth rates determined by the maximum concentration and mode fitting methods. The statistical results valid the potential of applying the proposed method on different sites, which will improve the automatic level for NPF events detection and analysis. Furthermore, the proposed automatic NPF event analysis method provides more consistent results compared with human-made analysis, especially when long-term data series are analyzed and statistically comparisons between different sites are needed for event characteristics such as the start and end times, thereby saving time and effort of scientists studying NPF events.



# 1 Introduction

Atmospheric aerosols have profound impacts on air quality, human health, ecosystem, weather, and climate (Asmi et al., 2011a; Hirsikko et al., 2011; Joutsensaari et al., 2018; Chu et al., 2019; Lee et al., 2019). New particle formation (NPF) is an important source of atmospheric aerosols, which has been observed in a variety of locations in the world such as different types of forests, semi or heavily polluted cities, high-altitude sites, coastal sites, and polar regions (Kulmala et al., 2004; Kuang et al., 2010; Kulmala et al., 2012; Nieminen et al., 2018; Dada et al., 2018; Lee et al., 2019). In addition to the spatial scale, on the temporal scale, NPF events have also been observed in sites built long term ago (Dal Maso et al., 2005; Järvi et al., 2009; Asmi et al., 2011b) and newly built sites (Kerminen et al., 2018; Chu et al., 2019; Liu et al., 2020; Yan et al., 2021).

To analyze NPF events, the first step is to determine whether an NPF event has occurred or not (Kulmala et al., 2012). Previous studies on detecting NPF types can be roughly divided into three categories: vision-based, rule-based, and data-driven. Vision-based methods visually classify the NPF types day by day according to some criteria based on surface plots of the size distribution time series (Mäkelä et al., 2000; Dal Maso et al., 2005; Hirsikko et al., 2011). The advantage of vision-based methods is that experts can explicitly tell which region in a surface plot is thought of as the evidence of an NPF event, and the drawbacks of vision-based methods are labor-intensive, time-consuming, and the classification process is subject to human bias. Rule-based methods classify NPF types with several explicit steps where some thresholds on the particle number concentrations are used as prior knowledge (Kulmala et al., 2012; Dada et al., 2018). Rule-based methods can classify NPF types automatically, but the drawback of these methods is that the particle number concentrations can vary a lot between different environments, meaning that the prior knowledge used in one site may fail on other sites or complex situations. Data-driven methods utilize the measured particle number size distributions and annotated NPF types (labels) to establish a model which can identify NPF types. For instance, neural networks (NNs) have been used to classify NPF types no matter whether handcrafted features (Nanni et al., 2017) are used (Zaidan et al., 2018) or not (Joutsensaari et al., 2018). The advantages of data-driven or NN-based methods are that they do not need any specific threshold on particle number concentration and the classification process is automatic. However, annotated NPF labels are required to train the NNs, and since the label annotation process is subjective, the trained NNs also "learn" the biases in the labels, which impedes the application of NN-based methods to different sites. Considering the increasing number of global observation stations (Kulmala, 2018), an automatic NPF detection method that applies to NPF datasets collected in different sites is necessary.

Although not all NPF events show signs of growth (Dal Maso et al., 2005), or have the commonly known "banana" shape, in this work, we only focus on the regional (banana-type) NPF events which are the most common type of events observed globally and whose formation signature is the continuous formation and subsequent growth of nucleation mode (sub-25 nm) particles. We observe that there are some similarities between recognizing NPF events in surface plots and other objects in digital images. Taking cats as an example, no unique mathematical criterion or definition for NPF events or cats can be found. However, humans can easily distinguish whether an NPF event occurs in a surface plot or whether a cat occurs in a digital image in most cases. Inspired by this observation, we regard the banana-type NPF events as a special kind of object, and thus the object detection techniques for detecting cats can be used to detect the banana-shaped NPF events. For simplicity,





we use *NPF images* to represent surface plots without axes. Though surface plots have clear physical meanings, we can apply different image transformations on NPF images without any restriction. In this study, we use an instance segmentation method called Mask R-CNN (He et al., 2017), a deep learning model, to localize the NPF events by predicting a mask that can cover
the spatial layout (the banana shape) of each NPF event. In other words, we try to answer the NPF classification problem by directly localizing the visual signature of NPF events. Since Mask R-CNN only focuses on the banana shape that has been observed globally, it can be used on datasets collected from different sites automatically. For more information about object detection and instance segmentation, please refer to Appendix A.

To verify the generality of the presented method, we test the Mask R-CNN model on three SMEAR stations (Station for
Measuring Ecosystem and Atmospheric Relations I, II, and III) in Finland and one station located in San Pietro Capofiume at the Po Valley basin in Italy (SPC station). The datasets collected in the four stations sum up approximately 73 years of measurements. Besides the classification problem, the accurate location of events makes it easier to determine the growth rates, start times, and end times automatically. Our code is released to test on datasets collected in other sites and facilitate future research. Our aims in this study are (1) to automatically localize the globally observed visual signature (banana shape) for
regional NPF events, which can identify NPF types (events occurs or not, especially for the strongest events), and determine the growth rates, start times, and end times, (2) to investigate the statistical characteristics of growth rates, start times, and end times for the strongest NPF events for the three SMEAR stations in Finland and the SPC station in Italy.

## 2   Materials and methods

### 2.1   Measurement sites

We utilized aerosol size distribution data from three observation sites in Finland and one in Italy. All the sites operated similar instrumentation and the observations followed guidelines set by the ACTRIS on in situ aerosol number size distribution measurements (Wiedensohler et al., 2012). The observation sites and instruments are shortly described below.

The SMEAR I station is located at the Värriö Subarctic Research Station of the University of Helsinki (67°46′N, 29°36′E, 390 m a.s.l.) in northern Finland. The station is surrounded by 70-year-old Scots pine (Pinus sylvestris) boreal forest at Koto-
vaara hill, while some small lakes and mires exist in valleys 60 m lower and more than 1 km far. The measurements of particle number size distribution started in 1997 in SMEAR I station. For more details about the site and measurements, please refer to Vana et al. (2016), Kyrö et al. (2014), and Hari et al. (1994). The analyzed particle number size distribution dataset collected in Värriö covers 8189 days from 10 December 1997 until 14 January 2021 (8436 days in total and the days with missing data were omitted from this study).

The SMEAR II station is located in Hyytiälä Forestry Research Station of the University of Helsinki in central Finland (61°51′N, 24°17′E, 130 m a.s.l.), within pine dominated boreal forest with some deciduous birch (Betula pubescens) and aspen (Populus tremuloides) trees. Comprehensive measurements including particle, radiation, gas, meteorological and complementary data have been measured for more than 20 years (Hari and Kulmala, 2005; Dada et al., 2017, 2018). The location is considered as a semi-clean boreal forest environment according to the level of anthropogenic pollutants (Nieminen et al., 2015;





Dada et al., 2018; Zaidan et al., 2018). A detailed overview of the site and measurements can be found in Hari et al. (2013). The analyzed particle number size distribution dataset collected in Hyytiälä covers 8642 days from 31 January 1996 until 21 January 2020 (8756 days in total).

The SMEAR III station is located in Kumpula campus of the University of Helsinki in southern Finland (60°12′N, 24°58′E, 26 m a.s.l.). The station has accumulated approximately 17 years of measurements such as air pollution, meteorological and

turbulent exchange (Järvi et al., 2009). The location is located within urban environment surrounded both by campus buildings, busy streets, open bedrock, and parklands of deciduous forest, such as birch, aspen, and maple (Acer pseudoplatanus). For more details about the site and measurements, please refer to Järvi et al. (2009) and Dada et al. (2020b). The analyzed particle number size distribution dataset collected in Kumpula covers 5775 days from 1 January 2005 until 14 January 2021 (5857 days in total).

The San Pietro Capofiume measurement station (SPC station) is located in a rural area (44°39′N, 11°37′E, 11 m a.s.l) in Po Valley, which is the largest industrial, trading, and agricultural area in Italy (Joutsensaari et al., 2018). The particle number size distribution measurements started from March 2002 and were carried out continuously, expect for occasional system malfunctions, until 2017. A detailed overview of the site and measurements can be found in Joutsensaari et al. (2018). The analyzed particle number size distribution dataset collected in SPC covers 4177 days from 24 March 2002 until 16 May 2017

(5534 days in total).

The aerosol particle number size distributions were measured by differential mobility particle sizer (DMPS) (Aalto et al., 2001) at all four stations (Fig. 1). The particle number size distribution datasets collected from the four stations are termed as Värriö dataset, Hyytiälä dataset, Kumpula dataset, and SPC dataset. The DMPS systems installed in different stations have different detection ranges for particle sizes, and particle sizes ranging from 3 to 1000 nm are considered in this work. Note that

the detected particle size does not have to reach 1000 nm for all DMPS systems.

## 2.2   NPF types

According to the guidelines reported in previous studies, the particle number size distributions can be classified into six different types (Dal Maso et al., 2005; Kulmala et al., 2012; Joutsensaari et al., 2018):

- *Class Ia events*. Ia-type events show clear and strong formation of small particles (especially 3–6 nm), with little or no
pre-existing particles in the smallest size ranges (Fig. 2a).

- *Class Ib events*. Ib-type events show the same behavior as class Ia but with less clarity (Fig. 2b).

- *Class II events*. II-type events do not show clear evidence for observing the growth. That is, the growth rate cannot be determined without a large uncertainty (Fig. 2c).

- *Class Non-Event* (NE). NE does not show any evidence for new particle formation in the nucleation particle size range
(Fig. 2d).

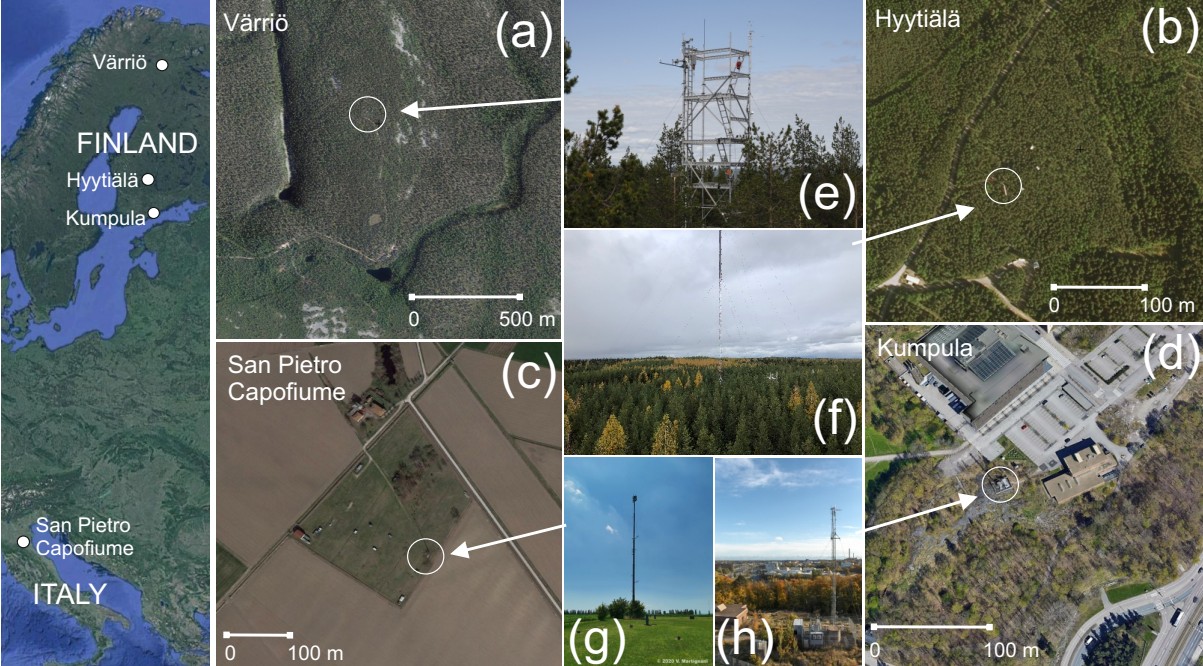

**Figure 1.** Värriö, Hyytiälä, Kumpula, and San Pietro Capofiume stations. Image courtesies: (a) and (b): Kansalaisen karttapaikka of National Land Survey of Finland (https://asiointi.maanmittauslaitos.fi/karttapaikka/), (c) and (d): © GoogleEarth, (e): Matias Uusinoka, 2021, (f): Petri Pellikka, 2019, (g): V. Martignani, 2020, (h): INAR, n.a.

- *Class Undefined* (Undef). Undef is a type that is difficult to be classified as events or NEs since some but not all features for events can be seen (Fig. 2e).

- *Class Bad-Data* (BD). BD type is caused by instrument malfunction. Generally, missing data or the particle concentrations are too high or too low can be observed (Fig. 2f).

Figure 2 shows the example surface plots for different NPF types. The banana shape can be seen clearly for Ia-type and Ib-type NPF events because they are so consistent throughout the day and are little influenced by local wind fields. Ia-type and Ib-type NPF events are usually connected with phenomena happening at large (regional) spatial scales. However, for II-type NPF events, interruptions in surface plots are often associated with more local sources of variability. The banana shape is not very clear for II-type NPF events and can be observed even in some Undef types.

**2.3   Mask R-CNN**

In order to fill the research gap mentioned in the introduction section, we used an object instance segmentation technique called Mask R-CNN, which can accurately localize an NPF event's spatial layout. Mask R-CNN extends the object detection method Faster R-CNN (Ren et al., 2016) by adding a new branch for generating segmentation masks of objects (He et al., 2017). Both





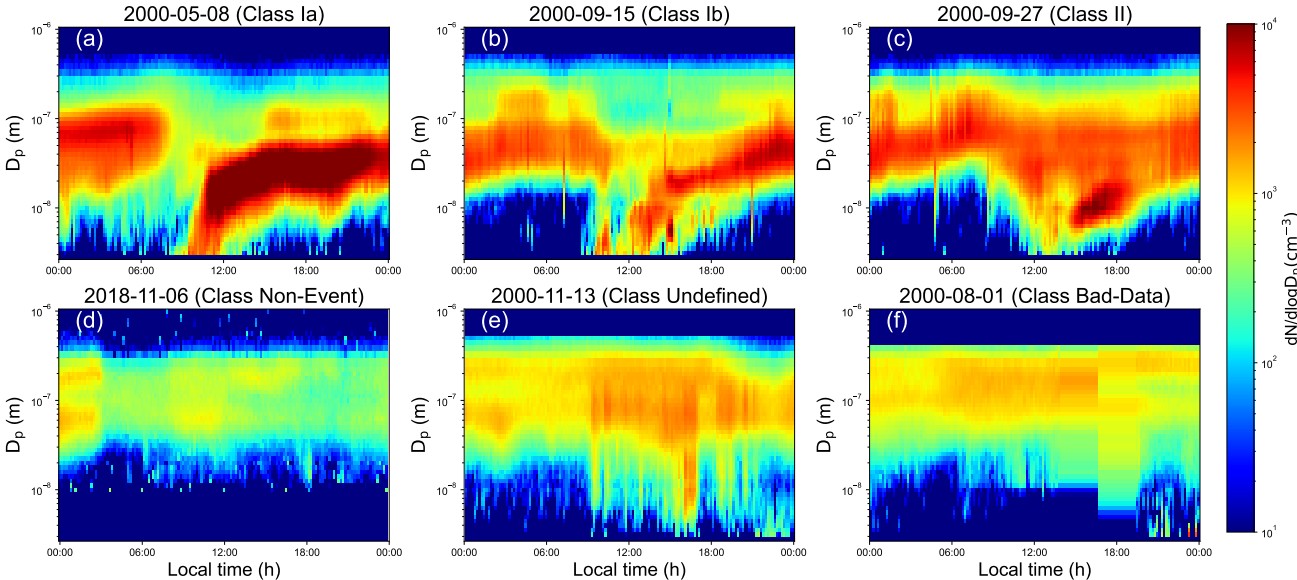

**Figure 2.** Example surface plots in the Hyytiälä dataset for different new particle formation types. Dates of measurements are used as titles for subplots and class names are in brackets.

the Mask R-CNN and Faster R-CNN models are derived from the Regions with CNN features (R-CNN) model (Girshick et al.,

2014), where CNN means convolutional neural networks. The architecture of Mask R-CNN is shown in Fig. 3. Similar to Joutsensaari et al. (2018), we fine-tuned the Mask R-CNN model which had been pre-trained on the Microsoft COCO dataset (Lin et al., 2014) with only 358 annotated masks. These 358 masks were created through a labeling tool *LabelMe* (Russell et al., 2008), and were from 358 NPF images (78 Ia-type, 202 Ib-type, and 78 II-type). The 358 NPF images were generated from the Hyytiälä dataset, and the period was from 1996 to 2003. During training, 300 NPF images with masks were randomly selected

as the training set, and the rest 58 NPF images with masks were the validation set (Fig. 4). The learning rate was $5 \times 10^{-3}$, and decreased every 3 epochs with a factor of 0.10. The stochastic gradient descent optimizer was used. We used weight decay of $5 \times 10^{-4}$ and momentum of 0.90. The Mask R-CNN model was fine-tuned for 10 epochs. All the NPF images and masks were resampled to $256 \times 256$ pixels, and with an NVIDIA V100 GPU, the training process lasted around five minutes. Code and more results are available at https://github.com/cvvsu/maskNPF.git.

Since Mask R-CNN only focuses on the banana shape, some regions in NPF images that are not events can also be localized, resulting in more than one mask that can be detected for one NPF image (Fig. 3). For each mask, there is an objectiveness score in [0, 1] showing the probability of an event occurrence. In addition to the objectiveness score, a bounding box is also obtained.

    Assuming the time resolution of DMPS systems are 10 minutes and there are 52 samples for particle sizes ranging from 3

to 1000 nm, the recorded particle number size distribution for one day is a data matrix with the shape of $52 \times 144$ (3 to 1000



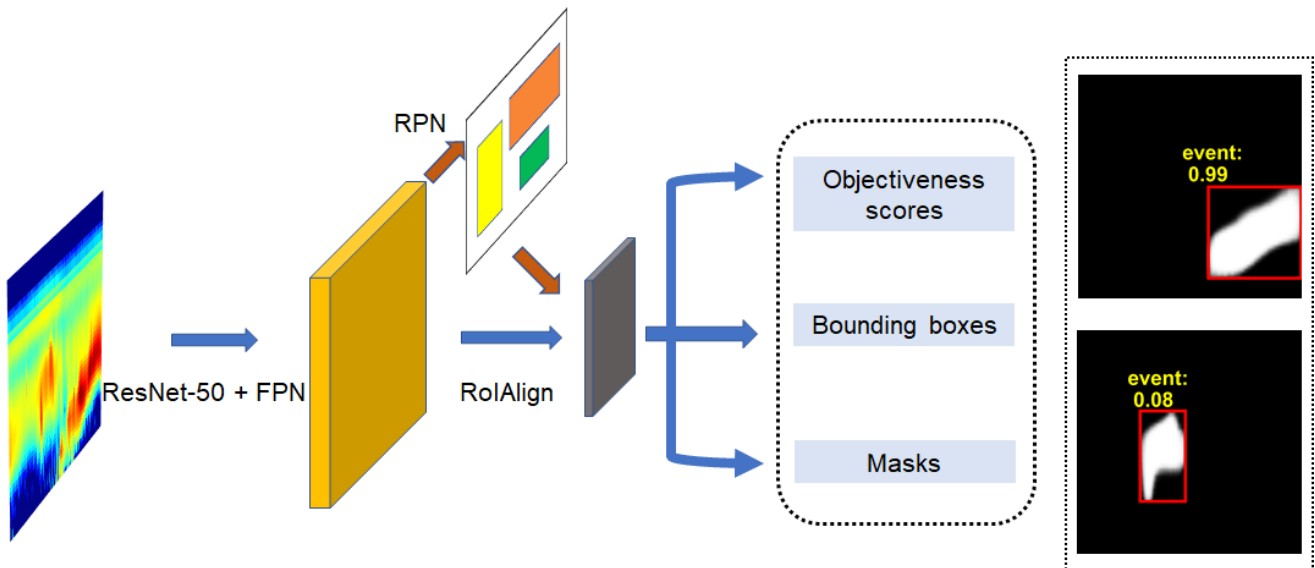

**Figure 3.** The architecture of Mask R-CNN. ResNet-50 is ResNet (He et al., 2016a, b) with 50 layers. FPN is the feature pyramid network (Lin et al., 2017). RPN is the region proposal network (Ren et al., 2016). RoIAlign is the RoIAlign layer that properly aligning the features (He et al., 2017).

nm from the bottom row to the top row and 0 o'clock to 24 o'clock from the first column to the last column). We resampled the predicted masks to the size of $52 \times 144$, aligning to the shapes of collected data (Fig. 5).

The value of a pixel in a mask represents the probability of the pixel belongs to an event. For each predicted mask, it was binarized at a threshold of 0.50 (He et al., 2017). The left and right edges of bounding boxes determine the start and end times, respectively. The bottom and upper edges of bounding boxes automatically provide a size window that covers the related NPF event (Fig. 3 and Fig. 5).

## 2.4 Growth rate

The particle growth rate (GR) is the rate of change for a given particle:

$$\text{GR} = \frac{dD_p}{dt} = \frac{\Delta D_p}{\Delta t} = \frac{D_{p2} - D_{p1}}{t_2 - t_1}, \tag{1}$$

where $D_{p2}$ and $D_{p1}$ is the particle diameters at times $t_2$ and $t_1$, respectively.

The maximum concentration method and log-normal distribution function (mode fitting) method are two widely used methods to calculate the growth rate (GR) for an NPF event (Kulmala et al., 2012; Dada et al., 2020a). The GRs determined by these two methods have the same order and seasonal variations (Dal Maso et al., 2005; Hirsikko et al., 2005; Yli-Juuti et al., 2011). Since the localization of the NPF events can be detected, we can accordingly calculate the GR of an NPF event automatically using the maximum concentration method. We used the random sample consensus (RANSAC) algorithm (Choi et al., 2009)





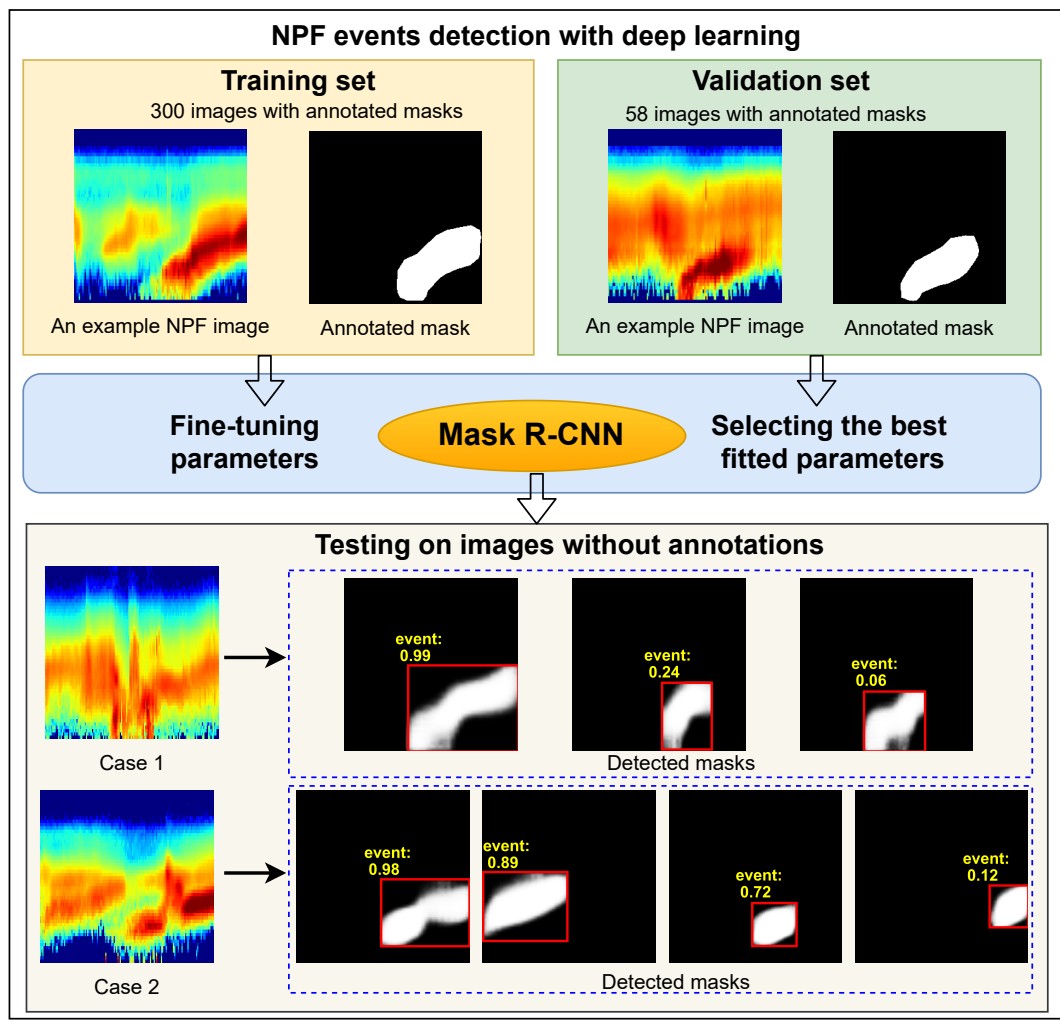

**Figure 4.** The flowchart of new particle formation events detection via Mask R-CNN.

instead of ordinary least square fitting to determine GRs. Compared to ordinary least square fitting, the RANSAC algorithm is robust to outliers. In addition to GRs, the predicted masks can also be used to analyze the characters of start times and end times of the strongest NPF events.

## 3 Results and discussion

### 3.1 Classification results

According to the classification results on the Hyytiälä dataset (Table 1), changing the threshold of objectiveness score does not affect the Ia and Ib types. However, on the SPC dataset, different thresholds have a big effect on the classification accuracy of



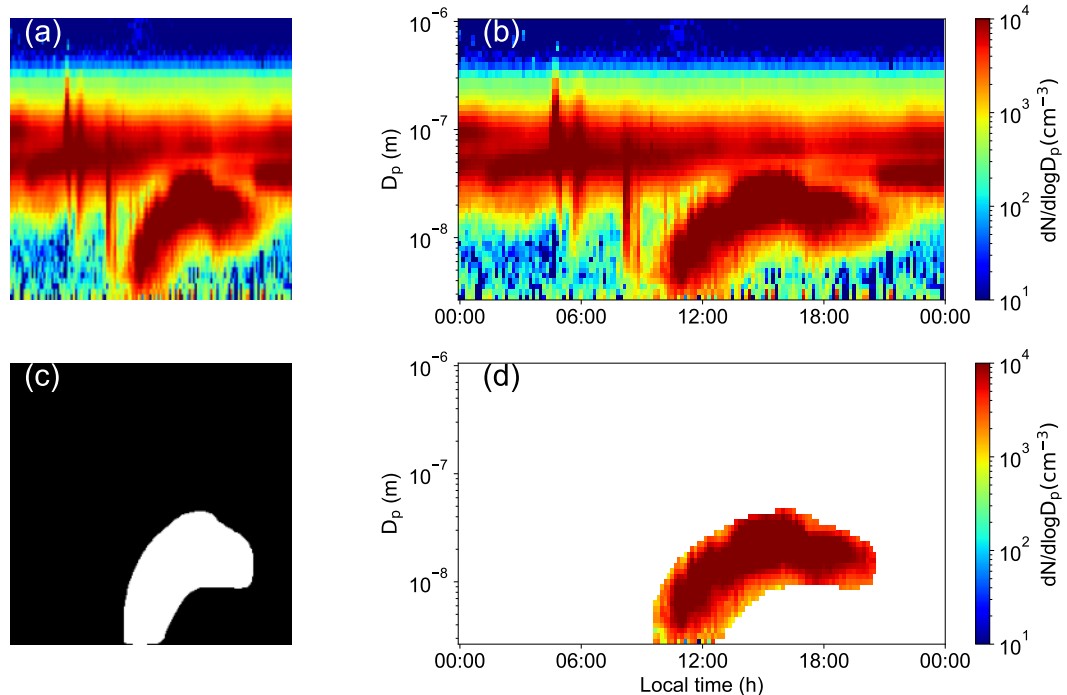

**Figure 5.** Example surface plot with the aligned mask. (a) An NPF image. (b) Related surface plot of (a). (c) Detected mask of (a). (d) Surface plot with the aligned mask.

**Table 1.** Classification results for NPF events of Ia, Ib, and II types on the Hyytiälä dataset (8642 days). Numbers in brackets are days for each class annotated by experts. Objectiveness score $\geq 0.00$ means that no threshold is applied and all the detected days are seen as event days.

| Objectiveness scores | Ia (137) | Accuracy for Ia (%) | Ib (722) | Accuracy for Ib (%) | II (1031) | Accuracy for II (%) | #Detected NPF days |
|---|---|---|---|---|---|---|---|
| $\geq 0.90$ | 137 | 100.00 | 688 | 95.29 | 690 | 66.93 | 1972 |
| $\geq 0.80$ | 137 | 100.00 | 703 | 97.37 | 759 | 73.62 | 2259 |
| $\geq 0.60$ | 137 | 100.00 | 706 | 97.78 | 829 | 80.41 | 2583 |
| $\geq 0.40$ | 137 | 100.00 | 711 | 98.48 | 880 | 85.35 | 2899 |
| $\geq 0.20$ | 137 | 100.00 | 715 | 99.03 | 922 | 89.43 | 3359 |
| $\geq 0.00$ | 137 | 100.00 | 717 | 99.31 | 967 | 93.79 | 4133 |

Ia and Ib types (Table 2). Since the Mask R-CNN model was trained on the masks derived from the Hyytiälä dataset, it did not contain any information about the SPC dataset, resulting in unstable classification accuracy when changing the threshold.





**Table 2.** Classification results for NPF events of Ia, Ib, and II types on the SPC dataset (4177 days) without annotations. Numbers in brackets are days for each class annotated by experts.

| Objectiveness scores | Ia (269) | Accuracy for Ia (%) | Ib (431) | Accuracy for Ib (%) | II (619) | Accuracy for II (%) | #Detected NPF days |
|---|---|---|---|---|---|---|---|
| ≥0.90 | 179 | 66.54 | 208 | 48.26 | 111 | 17.93 | 549 |
| ≥0.80 | 210 | 78.07 | 275 | 63.81 | 159 | 25.69 | 723 |
| ≥0.60 | 241 | 89.59 | 329 | 76.33 | 228 | 36.83 | 926 |
| ≥0.40 | 249 | 92.57 | 357 | 82.83 | 280 | 45.23 | 1065 |
| ≥0.20 | 255 | 94.80 | 379 | 87.94 | 347 | 56.06 | 1435 |
| ≥0.00 | 262 | 97.40 | 402 | 93.27 | 448 | 72.37 | 2000 |

According to the classification results shown in Table 1 and Table 2, there is a trade-off between the classification accuracy of NPF events and the number of misclassified days, which is controlled by the threshold. Re-training the Mask R-CNN model on masks derived from the SPC dataset may improve the classification accuracy on the SPC dataset and mask the classification results stable independent of the chosen threshold. We did not re-train the Mask R-CNN model to demonstrate the generality of our method (Table 2). Once a small threshold such as 0.20 for the objectiveness score is selected, on the SPC dataset and without annotated masks or class labels, the classification accuracy for Ia-type NPF events is 94.80%, for Ib-type NPF events is 87.94%, and for a combination of Ia-type and Ib-type NPF events is 90.57% (Table 2), which are higher than the results reported in Joutsensaari et al. (2018), where an NN-based method was applied. The classification results on the SPC dataset demonstrate the idea that regarding the banana shape in NPF images as a special object is reasonable.

According to the classification results of the four datasets, scientists who are only concerned about identifying Ia and Ib event types, this method will save them plenty of time and effort. Since the II-type events usually do not present a clear banana shape in the NPF images, the Mask R-CNN model fails to find some of these NPF events. However, detection results of Mask R-CNN can be used as auxiliary information to help determine the II types for scientists. Furthermore, the detection process is consistent compared with human-made determination.

### 3.2 Growth rate

In this study, we show that combined with the detected masks, the maximum concentration method can be used to calculate the GRs automatically (Fig. 5 and Fig. 6). If not specified, we only focus on determining the GRs, start times, and end times for the strongest NPF events.

Daytime hours between 6:00 and 18:00 (local time) were used for the traditional maximum concentration method to calculate the GRs. However, when the prior is not satisfied or particle burst presents in the surface plots, scientists need to select the start and end times manually. With the detected masks, the proposed method can automatically determine the time window (left and right edges of the bounding boxes, Fig. 3 and Fig. 5), and there is no need to manually adjust the start and end times. Usually,





different size windows were applied to calculate GRs, and we selected the 3–25 nm as the size range for GR calculation (Fig. 6). However, other size ranges are also possible, and for more information, please refer to our code. To avoid confusion, the maximum concentration and mode fitting methods are termed as traditional methods in this work.

As shown in Fig. 7, an obvious downtrend of GRs for the SPC station can be seen, and the medians of GRs for the SPC station are the highest than that for the other stations in most of the years, which is the same with the GRs determined by the traditional methods. The traditionally determined GRs of the SPC dataset utilized two different methods: from 24 March 2002 to 18 June 2011, the maximum concentration method was used, and from 19 June 2011 to 14 August 2017, the mode fitting method was applied. The medians of GRs for the Kumpula station is greater than that for Värriö and Hyytiälä stations but smaller than

that for the SPC station in most of the years, which is consistent with the observation that the GRs are highly related to the local pollution levels (Kulmala et al., 2005; Hamed et al., 2007). The Pearson correlation coefficients between traditionally and automatically determined GRs are 0.59 and 0.53 for the Hyytiälä and SPC stations, respectively. The traditionally determined GRs of the Hyytiälä station was calculated by the mode fitting method, which further verified that the GRs determined by the maximum concentration and mode fitting methods should have the same order and variations (Yli-Juuti et al., 2011). The

statistical results of GRs indicate the potential to utilize the automatic method to calculate GRs. Additionally, determining the GRs automatically leads to consistent results and gets rid of human errors.

### 3.3   Start time and end time

In addition to the GR, with the detected mask, the start time and end time of an event can also be determined automatically, which are only reported in very few publications (Kerminen et al., 2018; Dada et al., 2018). Figure 8 shows the start and end

times for the NPF events for different datasets. For the SPC dataset, the automatic method summarized the start times for events that occurred from 2002 to 2017, and the human-annotated results summarized the start times for events that occurred from 2011 to 2017. However, the histograms of the start times and end times determined by different methods show similar shapes (Fig. 8), illustrating the validity of the automatic method. Considering the end time of an event is difficult to determine in some cases, the end time of the NPF event cannot be identified as clearly as the start time.

Generally, the histograms of the start times for four datasets are bell-shaped, which may be controlled by normal distributions (Fig. 8). The histograms of end times for the SPC station also show the bell shape, but there is more than one peak in the histograms of end times for Värriö and Hyytiälä stations (Fig. 8). For NPF events that last for more than one day, interactions between particles in the two days lead to the end times being much more difficult to determine.

    The event durations for the NPF events for SPC station are generally shorter than those for Värriö, Hyytiälä, and Kumpula

stations (Fig. 8 and Table 3). The possible reason is that the atmospheric environment for the SPC station is much more polluted compared to the three SMEAR stations in Finland, making the events last for shorter times. The events in the Värriö and Hyytiälä stations have similar median durations, followed by Kumpula and SPC stations, possibly indicating that the atmospheric environment is less polluted in Värriö and Hyytiälä stations, then in the Kumpula station, and most polluted in the SPC station. Another possible reason is that Spring has the most frequent events and all stations other than SPC are higher in

latitude and thus have longer sunlight hours during Spring.





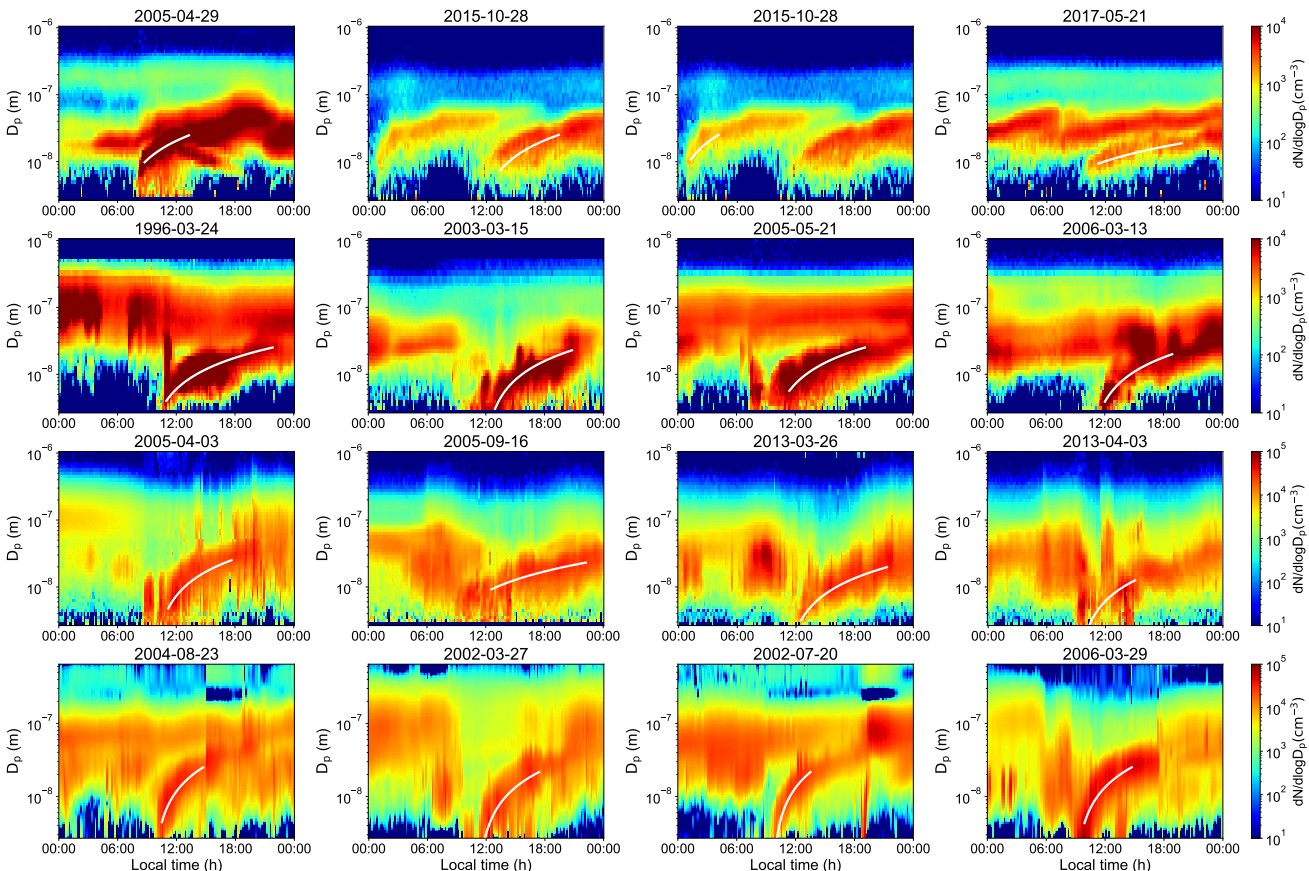

**Figure 6.** Examples of the automatically determined growth rates by the proposed method. From the top row to the bottom row, each row shows the examples (white curves) for Värriö, Hyytiälä, Kumpula, and SPC datasets, respectively. The date is shown as the title for each example.

The median start time is almost the same for the Hyytiälä and Kumpula stations (Table 3), which is consistent with that these two stations are located closely and further verifies that the intensity of solar radiation reaching the Earth's surface seems to be the most important factor affecting whether an NPF event occurs or not (Kerminen et al., 2018).

### 3.4 Advantages, limitations, and future studies

There are four major advantages of using the Mask R-CNN model (or other instance segmentation models) to detect NPF events. First, the process is simple, automatic, and straightforward. Second, the Mask R-CNN model can explicitly output masks for banana-shaped events, making the calculation of growth rates convenient together with the determination of start and end times. Third, the Mask R-CNN model can be used for datasets collected in different sites. For instance, the model trained on the masks from the Hyytiälä dataset works well on the SPC dataset. Fourth, the Mask R-CNN model is insensitive



**Figure 7.** Comparison of growth rates calculated by different methods. GR-T means that growth rates are determined by the traditional methods (manually selecting the start and end times when necessary), and GR-P means that growth rates are determined by the proposed automatic method. R is the Pearson correlation coefficient between GRs calculated by different methods. The density scatter plots in the bottom row show the ranges that the growth rates are usually located in.

to the sizes and aspect ratios of the input NPF images since the model has already "seen" the related image transformations during training.





**Figure 8.** Start and end times for the NPF events for different stations. The times mentioned in the plots are local times.

One limitation of using the Mask R-CNN model is that some days that are not NPF event days will be misclassified as event days. Therefore, for scientists focusing on the comparison between event and nonevent types, manual work is still required to select the misclassified days out. In this case, the Mask R-CNN model can only be used as an auxiliary tool.

The key to determine the correct start time, end time, and the GR for an event is that the detected mask can accurately depict the spatial layout of an NPF event. Since the Mask R-CNN model used in this study only trained on 358 annotated masks,





**Table 3.** Summary of start times, end times, and duration times for the NPF events for Värriö, Hyytiälä, Kumpula, and SPC datasets. A two-day unit was used for analysis. Hours more than 24 mean the related times in the next day.

|  |  | Värriö | Hyytiälä | Kumpula | SPC |
|---|---|---|---|---|---|
| Start time | Min | 00:29 | 06:29 | 04:49 | 04:56 |
|  | Max | 20:00 | 20:49 | 20:29 | 18:10 |
|  | Mean | 12:11 | 11:09 | 11:22 | 09:20 |
|  | Median | 11:55 | **10:58** | **11:04** | 09:16 |
| End time | Min | 10:30 | 11:39 | 09:30 | 12:25 |
|  | Max | 47:40 | 47:28 | 40:28 | 43:28 |
|  | Mean | 28:57 | 28:01 | 24:15 | 19:38 |
|  | Median | 30:00 | 28:58 | 24:10 | 18:47 |
| Duration | Min | 02:30 | 03:50 | 02:20 | 02:50 |
|  | Max | 37:30 | 37:10 | 29:18 | 33:10 |
|  | Mean | 16:46 | 16:52 | 12:53 | 10:18 |
|  | Median | 17:20 | 17:30 | 12:40 | 09:45 |

its generality may fail on some special observation stations. Thus, scientists need to re-train the Mask R-CNN model on their special datasets. However, there is no need to manually annotate masks again since some detected masks by our pre-trained model can be used as the annotated masks for the re-training.

## 4 Conclusions

To detect the NPF events automatically, we presented a method called Mask R-CNN for identifying the regional (banana-type) NPF events (especially the strongest events), and this method can also determine the growth rates, start times, and end times for events automatically. The method generalized well on different stations, and we tested the method on SMEAR I, II, and III (Värriö, Hyytiälä, and Kumpula, respectively) stations in Finland as well as the SPC station in Italy. All together approximately 73 years of measurements for datasets collected in the four stations were processed.

The proposed automatic method achieved the highest classification results for Ia-type and Ib-type events on the SPC station without any annotated information, showing the potential to apply the new method on other stations. The automatically determined growth rates by the new method are consistent with the manually calculated growth rates. The start times and end times determined by the new method illustrated that the start times may be controlled by normal distributions, but the end times presented more than one peak in their histograms for the Värriö and Hyytiälä stations.

This study promotes the development of the automatic detection of NPF events. The method presented can help to study the statistical properties of the strongest NPF events with minimal human participation, especially for the datasets acquired over a lengthy period. Besides a deeper understanding of the mechanisms of NPF, we will try to further improve the automatic level for NPF studies in the future.

*Code and data availability.* Code is available at https://github.com/cvvsu/maskNPF.git. Datasets collected in the three SMEAR stations are available at https://smear.avaa.csc.fi/. The dataset collected in San Pietro Capofiume station is available from Jorma Joutsensaari on request (Joutsensaari et al., 2018).

**Appendix A: Object detection and instance segmentation**

Object detection is one of the fundamental and challenging tasks in computer vision. Generally, some object detection tech-
niques focus on detecting different kinds of objects such as cats and cars, while others focus on specific scenarios such as face detection (Zou et al., 2019). With the development of deep learning, object detection achieves unprecedented improvements. The techniques can be roughly divided into one-stage detection such as single-shot multi-box detector (Liu et al., 2016) and two-stage detection such as Faster R-CNN (Ren et al., 2016). Usually, one-stage detection is much faster, while two-stage detection can achieve better detection accuracy. Instance segmentation, however, tries to delineate each distinct object of interest
in a more precise manner. In other words, instance segmentation segments an object according to its spatial layout. Compared with a bounding box which needs four corner positions to cover an object, an instance segmentation model needs to find all the pixels that belong in the object.

*Author contributions.* PS and PP: manuscript writing. PS, JJ, LD, MZ, TN, XL, and YW: data analysis and code. JJ and SD: SPC dataset and labels. TN, LD, and MZ: Hyytiälä dataset and labels. LD: Värriö dataset and labels. TP, MK, ST, TN, SD, XL, and YW: scientific
discussions. All authors comment on the raw manuscript.

*Competing interests.* The authors declare that they have no conflict of interest.

*Acknowledgements.* The work was supported by the MegaSense research programme of the University of Helsinki, the City of Helsinki Innovation Fund, Business Finland, Business Finland Project 6884/31/2018 MegaSense Smart City, the European Commission through the Urban Innovative Action Healthy Outdoor Premises for Everyone (project no. UIA03-240).We acknowledge also the following projects:
ACCC Flagship funded by the Academy of Finland (grant no. 337549), Academy professorship funded by the Academy of Finland (grant



no. 302958), and Academy of Finland PROFI funding (grant no. 311932). The authors wish to acknowledge CSC–IT Center for Science, Finland, for computational resources. The authors would like to thank Leone Tarozzi for the manually calculated growth rates.





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
