# Peer review of "New Particle Formation Events Detection with Mask R-CNN"

_Atmospheric Chemistry and Physics, 2021_

## Author Comment (AC1)

Dear editors and reviewers, thank you for your time, inspiring comments, and suggestions. The detailed revisions are listed below. Modified sentences are in blue. Code and pretrained models are now available at: https://github.com/cvvsu/maskNPF.git.

**Response to Referee #1:**

Reviewer 1 Comment 1 -- Major

In the section of 2.3 Mask R-CNN, description of the model is rather brief. The readability will be enhanced if authors provide a general description of convolutional neural networks and more details of the model, especially for ones not familiar with deep learning.

Thank you for your suggestion. We added a paragraph to further introduce the Mask R-CNN model in the modified manuscript as suggested (line 131-135):

The Mask R-CNN model can be seen as a learnable function $f_\theta$ that is parameterized by the learnable parameters $\theta$. That is,

$$Y = f_\theta(X) \tag{1}$$

where X is an input NPF image and Y contains three outputs: the class labels, bounding boxes, and masks. During training, the parameters are updated by reducing the losses between the output and annotated class labels, bounding boxes, and masks, leading to the best-fitted function $f_{\theta^*}$ (Girshick, 2015; Ren et al., 2016; He et al., 2017). The learned function $f_{\theta^*}$ is then applied to the test set to verify its generality.

Reviewer 1 Comment 2 -- Major

The Mask R-CNN model in this work was tuned with fixed training-validation ratio (300/58). Is testing set not necessary for the evolution of the model? Besides, the reason for choosing the training-validation ratio (300/58) and image size (256 × 256 pixels) should be explained, although it is mentioned that the Mask R-CNN model is insensitive to the sizes and aspect ratios of the input NPF images. (line 234-235)

Sorry for the confusion. Generalization capability is one of the most important criteria for evaluating the performance of a machine or deep learning model. Usually, the test set is used to demonstrate the generality of a model. Since the architecture or/and hyperparameters (e.g., learning rate) of a model need to be changed to have better

performance, it is better to split the dataset randomly into the training, validation, and test sets. The model is tuned on the validation set, and then the generality is verified on the test set. The model cannot see the test set during the training and validation phases.

In our case, we did not have an explicit test set to demonstrate the generality, but the classification results on different datasets had demonstrated the generality of the proposed method. For instance, the SPC dataset can be seen as a test set as the Mask R-CNN model has never seen the SPC dataset during training and validation phases.

Though the rule to split datasets into training, validation, and test sets is clear, ratios between the training, validation, and test sets are not fixed. The ratio of the test set (or the validation set) is usually in the ranges of 10% and 20%, but the ratio can be even smaller if the number of samples in a dataset is very large. In our case, the ratio between training and validation sets is approximately 84%:16%, which is a reasonable ratio.

The Mask R-CNN model is indeed insensitive to the sizes of the NPF images, but there is no need to draw the NPF images with different sizes. For simplicity, we utilized the NPF images with the size of 256 × 256 pixels as the input size. During the training phase, the Mask R-CNN model can transform the input images to different sizes and aspect ratios, making it possible to recognize objects in images with different sizes. An NPF image with the size of 256 × 256 or 128 ×128 will not qualitatively change the detection results. For instance, if we resample an NPF image on 15 September 2006 from the Hyytiälä dataset to different sizes, the detection results are almost the same (Table R1).

To avoid confusion, sentences were added or modified as below:

Data collected after 2003 in Hyytiälä, and datasets collected in Värriö, Kumpula, and SPC are the test sets. (line 144)

For practical usage, we can plot the NPF images with the size of 256 × 256 or 128 × 128 pixels for NPF events detection. (line 247-248)

**Table R1.** Detection results for different image sizes.

[Figure]

Reviewer 1 Comment 3 -- Major

As stated in line 123, there seems no distinct boundary between II-type NPF events and the Undef types, i.e. the overlapping between different types may occur. Hence, the uncertainty due to the overlapping should be discussed in the main text.

Thank you for the suggestion. It is true that some manually classified Undef-type days will be recognized as event days by the Mask R-CNN model (please see Table R2 and Table R3 below).

We added the discussion as follows (line 251-253):

As mentioned above, the Undef days are difficult to be classified with 100% certainty. Some manually classified Undef days are recognized as event days by the Mask R-CNN model (Table 1 and Table 2). These "misclassified" Undef days can be used as auxiliary information for scientists, in terms of classifying days to Undef type or not.

**Reviewer 1 Comment 4 -- Major**

In line 142, the value of objectiveness score is limited within the range of [0, 1]. However, how the value of objectiveness score corresponds to the exact classification type is not clear in the main text. This would originate from the characteristics of Mask R-CNN, and the authors should give an explanation.

Sorry for the confusion. There are three heads (objectiveness scores, bounding boxes, masks) in the output of the Mask R-CNN model as shown below.

[Figure]

**Figure R1.** The architecture of the Mask R-CNN model.

In our case, we have only two classes: banana and background (no banana). For a specific day, if there is no mask detected from its NPF image, then this day is not recognized as an event day by the Mask R-CNN model. Once a mask is detected, the Mask R-CNN model will output the objectiveness score to show its confidence (or the probability of being an event). We added one sentence in the modified manuscript (line 146-148):

Given a specific day, if no mask can be detected by the Mask R-CNN model, then this day will not be classified as an event day. On the other hand, if at least one mask is detected by the Mask R-CNN model, then this day will be recognized as an event day by the Mask R-CNN model.

Reviewer 1 Comment 5 -- Major

As shown in Table 1 & 2, the accuracy/performance of the model is dependent on the threshold of the objectiveness score, and the threshold of objectiveness score could vary dramatically when applying the model to other datasets. Can we simply set the threshold 0 to get the maximal accuracy? The authors should discuss the general criterion to choose the threshold of the objectiveness score.

Sorry for the confusion. Most all the events (especially the strongest ones) will indeed be detected if the objectiveness score is 0.0. Therefore, we recommend using 0.0 as the threshold for a one-year or two-year dataset. But for long-term datasets, it is better to set a small threshold such as 0.2 or 0.4 to save time and effort since the statistical characteristics may not change if a small portion of events is not included.

We added the related discussion below as suggested (248-250):

For short-term (one- or two-year) datasets, it is better to set the threshold as 0.0 for the objectiveness score to detect as many NPF events as possible, while for long-term datasets, a small threshold such as 0.2 or 0.4 will accelerate the detection and the statistical properties may not change if only a few event days are not included.

Reviewer 1 Comment 6 -- Major

The Conclusion section is too plain. The authors may want to summarize the novelty of this work here by comparing with previous works or point out the implications for future research.

Thank you for your suggestion. We modified the conclusion section as suggested (line 262-276):

With an increasing number of global observation stations, automatic NPF detection methods are required to speed up the NPF analysis process and minimize subjectivity caused by human-made analysis. To improve the automatic level of NPF detection, we presented a method called Mask R-CNN for identifying the regional (banana-type) NPF events (especially the strongest events), and the method can also be applied to determine the growth rates, start times, and end times for events automatically. The method generalized well on different stations, and we tested the method on SMEAR I, II, and III (Värriö, Hyytiälä, and Kumpula, respectively) stations in Finland as well as the SPC station in Italy. All together approximately 73 years of measurements for datasets collected in the four stations were processed.

The proposed automatic method achieved the highest classification results for Ia-type and Ib-type events on the SPC station without any annotated information, showing the potential to apply the new method on other stations. The automatically determined growth rates by

the new method are consistent with the manually calculated growth rates. The start times and end times determined by the new method illustrated that the start times may be controlled by normal distributions, but the end times presented more than one peak in their histograms for the Värriö and Hyytiälä stations.

In the future, the proposed method can be applied to datasets collected in different stations and over different time periods to produce comparable results, which will aid scientists in understanding the underlying mechanisms of NPF and assessing the impact of atmospheric aerosol particles on the climate.

**Reviewer 1 Comment 7 -- Minor**

The title is too closed to a previous study (Atmos. Chem. Phys., 18, 9597–9615, 2018) and suggested to be reorganized.

We modified the title to "*New Particle Formation Events Detection with Mask R-CNN*" as suggested.

**Reviewer 1 Comment 8 -- Minor**

In line 170, one may not identify the misclassification of NPF types directly from the Tables.

Sorry for the confusion. We modified Table 1 and Table 2 to:

**Table R2.** Classification results for NPF events of Ia, Ib, and II types on the Hyytiälä dataset (8642 days). Numbers in brackets are days for each class annotated by experts. Objectiveness score $\geq$ 0.00 means that no threshold is applied and all the detected days are seen as event days.

| Objectiveness scores | Ia (137) | Ib (722) | II (1031) | NE (1991) | Undef (3456) | BD (305) | Total (8642) |
|---|---|---|---|---|---|---|---|
| $\geq$0.90 | 137 | 688 | 690 | 18 | 419 | 20 | 1972 |
| $\geq$0.80 | 137 | 703 | 759 | 42 | 594 | 24 | 2259 |
| $\geq$0.60 | 137 | 706 | 829 | 65 | 815 | 31 | 2583 |
| $\geq$0.40 | 137 | 711 | 880 | 110 | 1023 | 38 | 2899 |
| $\geq$0.20 | 137 | 715 | 922 | 216 | 1312 | 57 | 3359 |
| $\geq$0.00 | 137 | 717 | 967 | 461 | 1770 | 81 | 4133 |

In Table 1, some Under, NE, and BD days are classified as event days by the Mask R-CNN model. We visualize these "misclassified" days in Appendix B (Fig. R2, Fig. R3, and Fig. R4) to help readers have a better understanding of the detection results. (line 186-188)

**Table R3.** Classification results for NPF events on the SPC dataset (4177 days) without annotations (labels). Numbers in brackets are days for each class annotated by experts.

| Objectiveness scores | Ia (269) | Ib (431) | II (619) | NE (1416) | Undef (841) | BD (601) | Total (4177) |
|---|---|---|---|---|---|---|---|
| ≥0.90 | 179 | 208 | 111 | 1 | 9 | 19 | 527 |
| ≥0.80 | 210 | 275 | 159 | 2 | 25 | 28 | 699 |
| ≥0.60 | 241 | 329 | 228 | 5 | 46 | 40 | 889 |
| ≥0.40 | 249 | 357 | 280 | 12 | 64 | 56 | 1018 |
| ≥0.20 | 255 | 379 | 347 | 101 | 141 | 107 | 1330 |
| ≥0.00 | 262 | 402 | 448 | 275 | 290 | 180 | 1857 |

[Figure]

**Figure R2.** "Misclassified" NE days detected by Mask R-CNN in the Hyytiälä dataset (the threshold for the objectiveness score is 0.9).

[Figure]

**Figure R3.** "Misclassified" Undef days detected by Mask R-CNN in the Hyytiälä dataset (the threshold for the objectiveness score is 0.9).

[Figure]

**Figure R4.** "Misclassified" BD days detected by Mask R-CNN in the Hyytiälä dataset (the threshold for the objectiveness score is 0.9).

Sentences were added to a new appendix section (Appendix B):

Example surface plots for the "Misclassified" NE, Undef, and BD days by the Mask R-CNN model (Fig. R2, Fig. R3, and Fig.R4). "Misclassified" means that these days were classified as NE, Undef, and BD days by scientists, while the Mask R-CNN model classified these days as event days. If the threshold for the objectiveness score is 0.9, then there are 18, 419, and 20 "misclassified" days for NE, Undef, and BD types, respectively. All the NE and BD days are shown in Fig. R2 and Fig. R4, but only the first 20 Undef days are shown in Fig. R3. These "misclassified" days can help readers understand the detection capability of the Mask R-CNN model.

Reviewer 1 Comment 9 -- Minor

According to the description in line 195-201, there may be some errors in the first panel of Figure 7.

Thank you for your careful comment. The figure should be:

[Figure]

**Figure R5.** Comparison of growth rates calculated by different methods. GR-T means that growth rates are determined by the traditional methods (manually selecting the start and end times when necessary), and GR-P means that growth rates are determined by the proposed automatic method. R is the Pearson correlation coefficient between GRs calculated by different methods. The density scatter plots in the bottom row show the ranges that the growth rates are usually located in.

**Response to Referee #2:**

Reviewer 2 Comment 1 -- Major

As the number of different ways of analyzing and identifying these events grows, so does the confusion of which one should be used. When presenting a new method, therefore, it would be good to have a comparison to other automated methods. Some methods are cited, but no comparisons are given, and therefore it is impossible to know which method to choose. I would very much like to see a an intercomparison between other similar automated methods. If this is not presented, this paper will just be another in a list of methods, and the user has no information on which one to use. Also, one method that is missing is Heintzenberg et al., 2007, doi: https://doi.org/10.1111/j.1600-0889.2007.00249.x There may also be others.

Sorry for the confusion.

In the introduction section of our manuscript, we illustrated that the previous classification methods could be roughly divided into three categories: vision-based, rule-based, and datadriven. We analyzed the advantages and disadvantages for each category (Table R4) and showed that why an automatic NPF classification method was still required.

**Table R4.** Advantages and disadvantages for different NPF events detection methods.

| Method | Advantages | Disadvantages | Related studies |
|---|---|---|---|
| Vision-based | Experts can explicitly tell which region is thought of as the evidence of an NPF event | Labor-intensive; time-consuming; classification results render subjective | Mäkelä et al., 2000; Dal Maso et al., 2005; Hirsikko et al., 2011 |
| Rule-based | Rule-based methods can classify events automatically | Prior knowledge is required. The prior works well on one station may fail on other stations. | Kulmala et al., 2012; Dada et al., 2018 |
| Data-driven | Data-driven methods | Data-driven methods need labels to train the model. However, the human-annotated labels themselves contain biases. | Zaidan et al., 2018; Joutsensaari et al., 2018 |

According to Joutsensaari et al. (2018), the previous automatic classification methods were not routinely used in NPF event analysis, and therefore, newly proposed methods were still coming out during the last few years (Dada et al., 2018; Joutsensaari et al., 2018; Zaidan et al., 2018). However, these recently proposed methods were not widely used, either. To further improve the automatic level for NPF analysis, we presented a new automatic classification method that is similar to the "vision-based" method by showing out the evidence (the banana shape) of an event.

Our proposed method avoids the disadvantages mentioned above and according to the classification results reported in Joutsensaari et al. (2018), our method achieved higher classification accuracy (Ia, Ib, and Ia + Ib) on the SPC dataset than the CNN-based method utilized in Joutsensaari et al. (2018). In addition, we did not use any annotated labels, which demonstrated the generality of our method.

For Heintzenberg's method (Heintzenberg et al., 2007), the authors stated that their method "*can only find events that it has been trained for*" in the "Discussion and conclusions" section, meaning that their method did not generalize well on unseen data. Plus, the authors mentioned that their method cannot "*find all three aforementioned types of events (strong, weak nucleation events, and events that are interrupted by short-term local meteorological process) at the same time*". Furthermore, they did not report their classification accuracy on their datasets, which makes their method difficult to compare with.

It is difficult to compare different methods if the methods are trained on private datasets and many parameters need to be manually determined in the methods. That is one of the reasons why we test our method on public datasets (SMEAR I, II, and III) and release our code: https://github.com/cvvsu/maskNPF.git.

> **Reviewer 2 Comment 2 -- Major**
>
> The actual phenomenon that is occurring is particle formation and growth in the atmosphere over a period over several hours, which is detectable by using a specific instrumentation and plotting this in a specific way. For example, the 'banana' images are a result of plotting the data in a specific way, which includes using a specific colormap with the logarithm of the log-normalized concentration density, which seems to be also hard-capped at some concentrations (1e4 in this case, based on the figures). For some situations, for example, the number concentration function value may well exceed the capping value, which changes the figure a lot. (Examples could be polluted megacities where even background concentrations can exceed 1e Also, some authors in the literature have used linear number distribution function values for plotting, and in these cases different features are highlighted. If, as I understand, this paper is very much focusing on analysing images instead of the actual data, then the various choices made in the image processing should be justified, and a section analysing the sensitivity of the model to this should be added. The authors note that the model is not sensitive to some image features such as aspect ratios, but from the text it is impossible to see whether other transformations (changes in colorscales, log/linear plotting, etc) affect the outcome. Of course, it would be highly interesting to do a double study, where the effect of such choices is studied on the human decision-making – here I think the automatic method could actually shine. But this might be outside of the scope of the study.

Sorry for the confusion. To avoid subjectiveness in the NPF analysis as much as possible, we did three jobs in this study:

1. Automatic detection of NPF event days (on NPF images or 'banana' images),
2. Automatic determination of start and end times for NPF events (on real data),
3. Automatic determination of growth rates for NPF events (on real data).

The Mark R-CNN model trained on the NPF images from the Hyytiälä station, which were from the plotting with $10^4$ as the maximum color scale. However, the model worked well on the NPF images from the SPC and Kumpula stations, which were from the plotting with $10^5$ as the maximum color scale (Fig. 6 in the manuscript).

**Table R5.** Detection results with different maximum color scales.

| Maximum color scale | NPF image | Detection results |
|---|---|---|
| $10^3$ |  |  |
| $10^4$ |  |  |
| $10^5$ |  |  |
| $10^6$ |  | None |

To visualize the surface plots, scientists in different stations will select different values (e.g., $10^4$, $10^5$, or $10^6$) as the maximum color scales. These values are chosen to make sure that the banana shapes can be clearly seen. For instance, researchers would not use $10^{100}$ as the maximum color scale since there would not be any banana shapes can be seen again. Specifically, a cat detection model will find a cat no matter the cat is white or black since colors will not affect the shape patterns in most cases. On the other hand, we cannot find a white cat in a white image.

We did a simple experiment to show that if the banana patterns are clear for scientists, then the Mask R-CNN model can also detect the pattern out. Different reasonable color scales do not qualitatively affect the detection results (Table R5).

Reviewer 2 Comment 3 -- Major

There is no discussion on whether the growth rates given by either method are related to the actual growth rate of the particles. There are different method for determining the growth rate (e.g. mode method, appearance time method etc.) as noted by the authors. These have specific physical meanings and their biases are at least to some extent known. The GR given by the present method seems to be just the maximum concentration method, which has some problems (for example, an aerosol that has the maximum inside a rather wide instrumental size bin will appear not to grow), I think that the authors should add discussion on the applicability of the results to estimate the actual physical parameters.

Sorry for the confusion.

The GRs were determined on real collected data, and they are the actual growth rates of the particles. Instead of utilizing the data collected in a whole day (Fig. R6 (b)), we only used the particle number size distribution in the banana region (Fig. R6 (d)).

To compare the advantages and disadvantages of different GR determination methods is not the objective of the current study. The maximum concentration method and mode fitting method have their advantages and disadvantages, respectively (Fig. R7). Therefore, these two methods are both utilized for GR calculation. In Fig. R7, the GRs in the first column are determined by the maximum concentration method, and the GRs in the second column are determined by the mode fitting method. In the first row, the maximum concentration method works well on different size bins, while the mode fitting method does not work well on small-size particles. In the second row, the GR determined by the mode fitting method seems more reasonable than the maximum concentration method.

[Figure]

**Figure R6.** Example surface plot with the aligned mask. (a) An NPF image (size: 256 × 256 pixels). (b) Related surface plot of (a). (c) Detected mask of (a) (size: 256 × 256 pixels). (d)Surface plot with the aligned mask.

[Figure]

(a) Maximum concentration

(b) Mode fitting

(c) Maximum concentration

(d) Mode fitting

**Figure R7.** GRs determined by the maximum concentration and mode fitting methods.

Reviewer 2 Comment 4 -- Minor

Last sentence of the abstract: What is meant by more consistent results? How is this defined? Consistent with what? If the results are consistent with manual results, how can they then be more consistent? This should be defined and clarified.

Sorry for the confusion. Since the manual classification process is subjective, different scientists will obtain different classification results. What is worse, a scientist may obtain different results at different times. As a result, the classification results are not consistent. For the Mask R-CNN model, it can alleviate the subjective processing as much as possible, leading to more consistent results. To avoid confusion, we modified the related sentence to:

Furthermore, the proposed automatic NPF event analysis method can minimize subjectivity compared with human-made analysis, especially when long-term data series are analyzed and statistically comparisons between different sites are needed for event characteristics such as the start and end times, thereby saving time and effort of scientists studying NPF events.

Reviewer 2 Comment 5 – Minor

line 212: "However, the histograms of the start times and end times determined by different methods show similar shapes (Fig. 8), illustrating the validity of the automatic method." Is the similarity of a histogram enough to validate the method? I think direct intercomparison of earlier data, and looking at the point-by-point difference would give a more robust way of looking at whether there is a bias or other error.

Sorry for the confusion. Differences (in minutes) between manually and automatically determined start times are shown in Fig. R8. Most of the differences are less than 1 hour. The summary of the differences is shown in Table R6. Note that the time resolution is 10 minutes for the particle number size distribution data.

**Table R6**. Summary of the differences between manually and automatically determined start times.

| Statistical indicator | Difference |
| --- | --- |
| Min | 00:00:00 |
| 25% | 00:00:51 |
| 50% | 00:10:29 |
| 75% | 00:20:35 |
| Max | 03:20:46 |
| Mean | 00:18:08 |
| Std | 00:27:25 |

[Figure]

**Figure R8.** Differences (in minutes) between manually and automatically determined start times.

To avoid confusion, we modified the sentence to (line 223-224):

However, the histograms of the start times and end times determined by different methods show similar shapes (Fig. 8).

Reviewer 2 Comment 6 – Minor

Figure 6: As one comparison is between traditional and automatic methods, could the GR given by the traditional methods be added to the figure too?

Sorry for the confusion. Utilizing the maximum concentration method to determine the growth rates (GRs), experts need to manually determine the start and end times of an event for some cases. There will be subjectiveness during the selection and as a result, there are no "standard" detection results, especially for events that do not have ideal banana shapes. However, our detection results can be reproduced directly: https://github.com/cvvsu/maskNPF/blob/main/demo.ipynb.

On the other hand, we compared our results with manually determined results in a statistical view (Fig. 7 in the manuscript).

References:

Dada, L., Chellapermal, R., Buenrostro Mazon, S., Paasonen, P., Lampilahti, J., E Manninen, H., Junninen, H., Petäjä, T., Kerminen, V. M. and Kulmala, M.: Refined classification and characterization of atmospheric new-particle formation events using air ions, Atmos. Chem. Phys., 18(24), 17883–17893, doi:10.5194/acp-18-17883-2018, 2018.

Dal Maso, M., Kulmala, M., Riipinen, I., Wagner, R., Hussein, T., Aalto, P., and Lehtinen, K.: Formation and growth of fresh atmospheric aerosols: eight years of aerosol size distribution data from SMEAR II, Hyytiälä, Finland, Boreal Env. Res., 10, 323–336, 2005.

Girshick, R.: Fast R-CNN, in: Proceedings of the IEEE International Conference on Computer Vision (ICCV), pp. 1440–1448, 2015.

Heintzenberg, J., Wehner, B. and Birmili, W.: 'How to find bananas in the atmospheric aerosol': new approach for analyzing atmospheric nucleation and growth events, Tellus B, 59(2), doi:10.3402/tellusb.v59i2.16988, 2007.

He, K., Gkioxari, G., Dollár, P., and Girshick, R.: Mask R-CNN, in: IEEE International Conference on Computer Vision (ICCV), pp. 2961– 2969, https://doi.org/10.1109/ICCV.2017.322, 2017.

Hirsikko, A., Nieminen, T., Gagné, S., Lehtipalo, K., Manninen, H. E., Ehn, M., Hõrrak, U., Kerminen, V.-M., Laakso, L., McMurry, P. H., Mirme, A., Mirme, S., Petäjä, T., Tammet, H., Vakkari, V., Vana, M., and Kulmala, M.: Atmospheric ions and nucleation: a review of observations, Atmos. Chem. Phys., 11, 767–798, https://doi.org/10.5194/acp-11-767-2011, 2011.

Joutsensaari, J., Ozon, M., Nieminen, T., Mikkonen, S., Lähivaara, T., Decesari, S., Facchini, M. C., Laaksonen, A. and Lehtinen, K. E. J.: Identification of new particle formation events with deep learning, Atmos. Chem. Phys., 18(13), 9597–9615, doi:10.5194/acp-18-9597-2018, 2018.

Kulmala, M., Petaja, T., Nieminen, T., Sipila, M., Manninen, H., Lehtipalo, K., Dal Maso, M., Aalto, P., Junninen, H., Paasonen, P., Riipinen, I., Lehtinen, K., Laaksonen, A., and Kerminen, V.-M.: Measurement of the nucleation of atmospheric aerosol particles, Nat. Protoc., 7, 1651–1667, https://doi.org/10.1038/nprot.2012.091, 2012.

Mäkelä, J. M., Maso, M. D., Pirjola, L., Keronen, P., Laakso, L., Kulmala, M., and Laaksonen, A.: Characteristics of the atmospheric particle formation events observed at a borel forest site in southern Finland, Boreal Env. Res., 5, 299–313, 2000.

Ren, S., He, K., Girshick, R., and Sun, J.: Faster R-CNN: towards real-time object detection with region proposal networks, IEEE Transactions on Pattern Analysis and Machine Intelligence (TPAMI), 39, 1137–1149, https://doi.org/10.1109/TPAMI.2016.2577031, 2016.

Zaidan, M. A., Haapasilta, V., Relan, R., Junninen, H., Aalto, P. P., Kulmala, M., Laurson, L. and Foster, A. S.: Predicting atmospheric particle formation days by Bayesian classification of the time series features, Tellus, Ser. B Chem. Phys. Meteorol., 70(1), 1–10, doi:10.1080/16000889.2018.1530031, 2018.